# Immunosuppressive Therapy Modifies Anti-Spike IgG Subclasses Distribution After Four Doses of mRNA Vaccination in a Cohort of Kidney Transplant Recipients

**DOI:** 10.3390/vaccines13020123

**Published:** 2025-01-25

**Authors:** Ignacio Juarez, Isabel Pérez-Flores, Arianne S. Aiffil Meneses, Ana Lopez-Gomez, Natividad Calvo Romero, Beatriz Rodríguez-Cubillo, María Angeles Moreno de la Higuera, Belén Peix-Jiménez, Raquel Gonzalez-Garcia, Beatriz Amorós-Pérez, Benigno Rivas-Pardo, Elvira Baos-Muñoz, Ana Arribi Vilela, Manuel Gómez Del Moral, Ana Isabel Sánchez-Fructuoso, Eduardo Martínez-Naves

**Affiliations:** 1Immunology Department, Complutense University of Madrid, 28040 Madrid, Spain; anlope22@ucm.es (A.L.-G.); beamoros@ucm.es (B.A.-P.); benignor@ucm.es (B.R.-P.); emnaves@ucm.es (E.M.-N.); 2Nephrology Department, Institute San Carlos for Medical Research (Instituto de Investigación Sanitaria del Hospital Clínico San Carlos (IdISSC)), San Carlos Clinical University Hospital, 28040 Madrid, Spain; perezfloresi@yahoo.es (I.P.-F.); naticalvo75@gmail.com (N.C.R.); brcubillo@gmail.com (B.R.-C.); sanchezfructuoso@gmail.com (A.I.S.-F.); 3Microbiology Department, Institute San Carlos for Medical Research (Instituto de Investigación Sanitaria del Hospital Clínico San Carlos (IdISSC)), San Carlos Clinical University Hospital, 28040 Madrid, Spain; 4Department of Cell Biology, Complutense University of Madrid, 28040 Madrid, Spain; mgomezm@med.ucm.es

**Keywords:** kidney transplant recipients, SARS-CoV-2 vaccine, serum neutralizing activity, IgG subclasses, COVID-19

## Abstract

**Background**: IgG4 is the least immunogenic subclass of IgG. Immunization with mRNA vaccines against SARS-CoV-2, unlike other vaccines, induces an increase in IgG4 against the spike protein in healthy populations. This study investigated whether immunosuppressive therapy affects the immune response, focusing on IgG subclass changes, to four doses of mRNA vaccine in kidney transplant recipients (KTRs). **Methods**: This study includes 146 KTRs and 23 dialysis patients (DPs) who received three mRNA-1273 vaccine doses and a BNT162b2 booster. We evaluated anti-spike IgG titers and subclasses, T-CD4+ and T-CD8+ cellular responses, and serum neutralizing activity (SNA). **Results**: At the fourth dose, 75.8% of COVID-19 naïve KTRs developed humoral and cellular responses (vs. 95.7% in DPs). There was a correlation between anti-spike IgG titers/subclasses and SNA (*p* < 0.001). IgG subclass kinetics after the third/fourth dose differed between COVID-19 naïve KTRs and DPs. Immunosuppressive therapy influenced IgG subclasses: mTOR inhibitors (mTORi) positively influenced IgG1 and IgG3 (*p* < 0.05), while mycophenolic acid negatively affected IgG1, IgG3, and IgG4 (*p* < 0.05). SNA is correlated with breakthrough infections after four doses of vaccine in KTRs. mTORi was the only factor associated with SNA > 65% in naïve KTRs [4.29 (1.21–15.17), *p* = 0.024]. **Conclusions**: KTRs show weaker cellular and humoral immune responses to mRNA vaccines and a class shift towards non-inflammatory anti-S IgG4 upon booster doses. IgG subclasses show a positive correlation with SNA and are influenced by immunosuppression. Increased SNA after four doses of vaccine is protective against infection. mTORi may benefit non-responding KTRs.

## 1. Introduction

Two mRNA vaccines from Pfizer-BioNTech (BNT162b2) and Moderna (mRNA-1273) were the first FDA-approved mRNA vaccines for immunization against the severe acute respiratory syndrome coronavirus 2 (SARS-CoV-2). Both showed high initial efficacy of around 90% in the general population [1,2], but we do not know the kinetics of this immunity in the medium and long term and the booster doses needed to maintain immunity. It is currently unknown how these new vaccine formats will affect the long-term B cell and Fc antibody response.

Some works have documented that, after three doses, immunization was superior in terms of humoral response and neutralizing capacity against several SARS-CoV-2 variants [3,4]. Knowledge of these data may be even more important in kidney transplant recipients (KTRs), as their immunosuppressed status results in a poorer and less durable response than in the general population [5].

Antibodies display multiple functions and features, and are fundamental to the control of viral infection. Neutralizing antibodies and IgG subclasses have been extensively studied as potential markers of disease severity and vaccine efficacy. Recent studies have shown that COVID-19 patients and those receiving vaccination with mRNA-based COVID-19 vaccines develop a robust IgG response in healthy adults, consisting mainly of IgG1 and IgG3 subclasses, with neutralizing activity against SARS-CoV-2 [6,7].

However, other authors have observed a significant increase in IgG4 levels after vaccination with the booster doses of the mRNA vaccines, contrary to other antigen expositions, such as adenoviral or inactivated vaccines, or after SARS-CoV-2 infection [8,9]. To the best of our knowledge, it is unknown whether this shift towards IgG4 production occurs when vaccinating individuals who are being treated with immunosuppressants.

IgG is the most prevalent immunoglobulin isotype in human serum and plays a pivotal role in the adaptive and innate immune responses. IgG subclasses have unique functional properties that are essential for host defense against infections [10,11]. These subclasses exhibit different affinities for Fc gamma receptors (FcγRs) and for C1q complement binding [12], which modulate their ability to trigger effector mechanisms such as antibody-dependent cellular cytotoxicity (ADCC) and antibody-dependent cellular phagocytosis (ADCP) [13,14].

IgG1 is the most prevalent subclass in serum and has a high affinity for FcγRI, FcγRIIa, and FcγRIIIa expressed on phagocytic and natural killer (NK) cells [15,16]. This subclass is associated with Th2-type responses and is involved in the clearance of extracellular pathogens by ADCC and complement activation [17]. IgG2 has a lower affinity for FcγRs but a higher affinity for C1q [18] and is critical for opsonization of encapsulated bacteria. IgG3 has the highest affinity for FcγRs and is the most potent activator of ADCC and complement activation, while IgG4 has the lowest affinity for FcγRs and C1q, is associated with immune tolerance, and is involved in the suppression of allergic and autoimmune reactions [12].

The kinetics of IgG subclasses may explain the variable humoral response observed in some individuals following vaccination. Investigating the dynamics of IgG subclasses is essential for understanding the mechanisms of protection against viral infection, especially in immunosuppressed populations. The aim of this study was to investigate the kinetics of anti-spike antibodies, serum neutralizing activity (SNA), and IgG1-IgG4 subclasses in a cohort of KTRs receiving four doses of mRNA vaccines, to analyze the effect of immunosuppressive therapy and to compare their response with that of another group of dialysis patients (DPs).

## 2. Methods

### 2.1. Study Design and Sample Collection

We conducted a prospective study involving a cohort comprising 146 KTRs who received three doses of mRNA-1273 vaccine at 0, 1, and 5 months and a booster dose of BNT162b2 vaccine at 11 months after the first dose. For comparison purposes, an additional cohort of 23 DPs who received the same vaccination regimen was included. Individuals who had previously been infected with SARS-CoV-2 or who became infected during the follow-up period were excluded from the initial vaccine effectiveness analysis.

Blood samples were collected two months after the fourth dose (P4). Previous sample data were obtained from samples collected in a previous study [5], corresponding to before vaccination (P0), 21 days after the second dose (P1), three months after the second dose (P2), and two months after the third (P3) and fourth dose (P4). Eight months after the fourth dose, we assessed patients who remained uninfected (naïve KTR) or had a breakthrough infection. The study design and sampling schedule are summarized in Figure 1.

### 2.2. Patient Selection

KTRs attending the outpatient Renal Transplant Unit between 1 March and 15 April 2021 and expressing a desire to be vaccinated were included in this study. Inclusion criteria included being over 18 years of age, having a history of kidney transplantation for at least six months, and consenting to participate in the study. Exclusion criteria included a history of malignancy, a solid organ transplantation other than kidney, primary immunodeficiency disease, previous allergic reactions to inactivated vaccines, and the presence of an unexplained fever above 37.5 °C or symptoms suggestive of infection.

The study was conducted in accordance with the ethical standards of the Declaration of Helsinki and was approved by the local ethics committee. Informed written consent was obtained from all participants before blood samples were taken.

The vaccination periods were carried out following the schedules and periods provided by the Spanish Ministry of Health, including the administration of a booster (4th dose) of the vaccination in individuals undergoing renal transplantation. The vaccination protocol was the same for all participants, including two initial doses of mRNA-1273, and a booster (third dose) of mRNA1237 was performed with 0.25 mL containing 50 μg of mRNA, i.e., half the dose of the initial regimen and a fourth dose of BNT162b2 vaccine (due to stock limitations).

Samples were taken according to the literature available at the time, although a standard of two months after the booster dose was established for the last samples, based on the results published after one year of analysis of immune responses in the general population.

### 2.3. Control Group Selection

We established a control group of non-immunocompromised individuals to compare the immune response to the vaccine. This control group consisted of 23 DPs (naïve to SARS-CoV-2 infection and with no breakthrough infection) who received the same vaccination schedule. Blood samples were collected from the DPs at equivalent time intervals after vaccination under the same conditions as the KTRs.

### 2.4. SARS-CoV-2 Serology Analysis

A quantitative SARS-CoV-2 IgG test (SARS-CoV-2 IgG II Quant; Abbott Diagnostics) was used according to the manufacturer’s instructions. Samples with AU/mL values ≥50 were considered positive for SARS-CoV-2 IgG antibodies. To identify individuals who had been infected with the SARS-CoV-2 before vaccination or between sample collections, we employed an ELISA assay that assessed the presence of anti-SARS-CoV-2 N-protein antibodies, as previously described [5].

The antibody values were converted into WHO binding antibody units (BAU)/mL using a conversion factor of 0.142 (BAU/mL = AU/mL × 0.142).

### 2.5. Assessment of Cell-Mediated Immunity

Peripheral blood mononuclear cells (PBMCs) were obtained by density gradient isolation, and cells were stimulated with peptides covering the immunodominant domain surface S-protein of SARS-CoV-2. T-cell responses were assessed by flow cytometry using a lower limit of 0.05% as the threshold for a positive cellular response. The reported frequency was obtained by subtracting the background from the negative control for each donor. Antibodies employed were anti-CD3-FITC (UCHT1), CD4-PE (OKT4), and CD8-PE/Cy7 (SK1), and anti-IFNγ-APC (B27) (Biolegend), following standard procedures for intracellular staining. The gating strategy and dot-plots are shown in Appendix A.

### 2.6. Assessment of Serum Neutralizing Activity of ACE2-RBD

To determine the serum neutralizing activity (SNA) against the ACE2–Spike (RBD) interaction, a competitive ELISA assay was performed as previously described [5]. Briefly, plates coated with recombinant RBD and sera samples were incubated with biotin-labeled ACE2 to assess the ACE2-RBD interaction, resulting in a percentage neutralization value.

### 2.7. Measurement of IgG Subclasses

An ELISA assay for RBD protein was used to measure IgG1, IgG2, IgG3, and IgG4 isotypes. Ninety-six-well plates were coated with SARS-CoV-2 RBD (2 μg/mL) for 16 h at 2 °C, sera samples (1:100 dilution) were incubated, and antibodies were detected using goat anti-human IgG1, IgG2, IgG3 and, IgG4 HRP-conjugated antibodies (SouthernBiotech, Birmingham, AL, USA). Data from these ELISAs are expressed as optical density (OD) at 450 nm.

To establish the cut-off of anti-S IgG subclasses, we used the value of the mean plus twice the standard deviation (Mean OD + 2 * SD, 95% CI) of the OD value at 430 nm of 8 pre-pandemic sera (PCR negative, anti-S and anti-N total IgG negative, and with no COVID-19 compatible symptoms) per ELISA plate. This methodology is based on the fact that 95.0% of the data are within 1.96 standard deviations of the mean. Therefore, values outside this range are unlikely to be due to random sampling and are significantly higher than the cut-off point value obtained for each IgG isotype. Values above the cut-offs were considered positive. The cut-offs were established as follows: IgG1: 0.0085; IgG2: 0.0075; IgG3: 0.0559; IgG4: 0.018.

### 2.8. Statistical Analysis

Quantitative data are presented as mean and standard deviation (SD) or median with interquartile range (IQR), while qualitative variables are expressed as absolute and relative frequencies. Categorical variables were compared using the chi-squared test, and continuous variables were analyzed using Student’s *t*-test or the Mann–Whitney U test, as appropriate. Repeated measures were compared using the Wilcoxon signed-rank test or the McNemar test. Spearman’s rho was used to assess the correlations between continuous variables. Logistic regression was used to assess factors associated with immune response, with factors showing a univariate association (*p*-value < 0.100) included in the final multivariate model. All calculations were performed using GraphPad Prism version 8.0 and SPSS version 25.0, with a significance level of *p* < 0.05 (two-tailed).

## 3. Results

### 3.1. Naïve Kidney Transplant Recipients Have a Lower Response Rate After Fourth Dose of mRNA Vaccine than Dialysis Patients

The demographic characteristics, immunosuppressive therapy, and laboratory findings of all patients are described in Table 1. Among the COVID-19 naïve KTRs, 75.8% (69/91) developed a humoral and cellular response after the fourth dose (vs. 95.7% in DPs, *p* < 0.05). Of the remaining 22 (24.2%) naïve KTRs, 8 patients developed only cell-mediated immunity, 11 developed only a humoral response, and only 3 had no immune response, either cellular or humoral.

In addition, the intensity of the humoral immune response after the fourth dose was also significantly higher in DPs than in KTRs: anti-spike IgG titers of 23,587 (14,275–33,931) vs. 3640 (279–15,940) AU/mL, *p* = 0.002; SNA 70.08% (49.14–85.96) vs. 29.23% (30.03–62.93), *p* = 0.006. However, no differences in the intensity of the cellular response were observed between DPs and KTRs who had a positive response: reactive CD4 T-cell 0.18 (0.03–0.49) vs. 0.28% (0.25–0.56), *p* = 0.535; reactive CD8 T-cell 0.24 (0.08–0.63) vs. 0.15 (0.01–0.48), *p* = 0.288.

### 3.2. Kinetics of Anti-Spike IgG Subclasses After Vaccination Against SARS-CoV-2 in Naïve Kidney Transplant and Dialysis Patient

In naïve KTRs, median levels of IgG1 were below the cut-off after P1 and it was increased only after the fourth dose (P4). IgG2 was slightly increased after P2 without further increase in subsequent doses. IgG3 was not detected above the cut-off, while median IgG4 levels were only detected after the fourth dose (P4) (see Table 2 and Appendix A).

In naïve DPs, a decrease in IgG1 levels was observed after the fourth dose compared to the third dose, and IgG4 was the most prevalent IgG subclass.

Except for IgG1, DPs developed higher levels of the different IgG subclasses than KTRs after four doses of vaccine (*p* < 0.001).

### 3.3. Effect of Immunosuppressive Therapy on the Kinetics of Anti-Spike IgG Subclasses

Significantly, immunosuppressive therapy had a notable effect on the levels of IgG subclasses in naïve KTRs. Mycophenolate (MPA) had a negative effect on IgG1, IgG3, and IgG4 subclasses after four doses of vaccine, whereas mTOR inhibitors (mTORi) had a positive correlation with pro-inflammatory subclasses (IgG1 and 3). KTRs treated with MPA exhibited lower levels of IgG1 [0.031 (0.003–0.193) vs. 0.127 (0.030–0.364), *p* = 0.047], IgG3 [0.011 (0.004–0.077) vs. 0.034 (0.020–0.100), *p* = 0.039], and IgG4 [0.014 (0.002–0.097) vs. 0.329 (0.069–0.516), *p* = 0.006] than untreated KTRs (Figure 2A). Conversely, mTORi treatment showed significantly higher levels of IgG1 [0.153 (0.050–0.267) vs. 0.023 (0.004–0.137), *p* = 0.014] and IgG3 [0.038 (0.020–0.121) vs. 0.010 (0.004–0.067), *p* = 0.024] than KTRs without mTORi (Figure 2B). Moreover, patients treated with mTORi had a significant increase in total anti-S IgG and SNA after the fourth dose (Appendix A).

No association was found between IgG subclasses and age, gender, or other immunosuppressive therapies.

### 3.4. Correlation of Anti-Spike IgG Titers and Subclasses with SNA. Factors Associated with SNA

A robust correlation was found between the titers of anti-spike IgG and SNA after the fourth dose in naïve KTRs (r = 0.823, *p* < 0.001). Moreover, a positive correlation was observed between the different IgG subclasses and SNA, with the strongest correlation noted with IgG1 (IgG1: r = 0.706, *p* < 0.001; IgG2: r = 0.358, *p* = 0.007; IgG3: r = 0.529, *p* < 0.001; IgG4: r = 0.502, *p* < 0.001) and with total IgG (r = 0.823 *p* < 0.0001) (Figure 3).

Finally, we examined potential factors associated with achieving SNA > 65% after the fourth dose, which represents the 75th percentile of the data set.

Only mTORi therapy was statistically associated with SNA in COVID-19 naïve KTRs in the multiple logistic regression analysis. Patients receiving this treatment were more than four times more likely to develop high SNA [4.29 (1.21–15.17), *p* = 0.024], as shown in Table 3.

### 3.5. Impact of Infection Before the Fourth Dose on the Vaccine-Induced Immune Response

Fifty-five KTRs (37.6%) developed either symptomatic or asymptomatic SARS-CoV-2 breakthrough infection after the booster dose. Seventeen patients had a history of COVID-19 prior to the first vaccine dose, eleven of whom met the criteria for severe COVID-19. Ten KTRs became infected after the second dose of vaccine, while twenty-eight were infected after the third dose (Figure 1). All patients who became infected after vaccination experienced mild or no symptoms.

These 55 infected KTRs exhibited a significant increase in anti-spike IgG titers after receiving the fourth dose, as evidenced in Table 4 (see Appendix A for a visual representation of the data). Furthermore, infected KTRs also showed a higher SNA than naïve KTRs [30.0 (7.9–62.9) vs. 74.4 (51.3–82.4)%, *p* < 0.001]. Interestingly, we found that infected KTRs had higher IgG titers for the different IgG subclasses than naïve patients, although this was statistically significant only for IgG3 [0.016 (0.006–0.077) vs. 0.098 (0.055–0.222), *p* < 0.001] and IgG4 [0.028 (0.003–0.214) vs. 0.155 (0.056–0.912), *p* = 0.003]. Regarding the cell-mediated immune response, infected KTRs also showed a higher proportion of reactive CD8 T-cells than their naïve counterparts (Table 4). No differences in the immune response were observed according to the severity of the infection.

### 3.6. Correlation Between Humoral Immunity and Incidence of Breakthrough Infection After the Fourth Vaccine Dose

During the six months following P4 (2 months after the fourth dose), 25 of the 91 naïve KTRs became infected. These 25 patients exhibited a weaker humoral immune response after four doses compared to those who remained uninfected, as evidenced by lower IgG titers and SNA percentage (Table 5).

Notably, KTRs who achieved an SNA greater than 65% after the fourth dose were almost four times less likely to become infected in the following six months (OR 3.63, 95%CI 0.91–14.55, *p* = 0.034).

Similar results were found for IgG subclasses. Infected KTRs had lower IgG subclass titers compared to non-infected patients, as shown in Table 5.

## 4. Discussion

Our results showed that KTRs have a high risk of breakthrough infection after the generally scheduled vaccination, even showing suboptimal seropositive protection after the third dose [5,19,20], which led to the recommendation of a fourth dose for these patients. Recent studies suggest that a fourth dose of the mRNA-based Pfizer-BioNTech or Moderna vaccines may offer short-term benefits to individuals with compromised immune systems, particularly kidney transplant recipients (KTRs), by improving immune responses. The World Health Organization has indicated that additional booster doses, such as a second booster, could enhance protection in these populations, although the overall benefit continues to be evaluated [21,22,23,24]. However, few studies have evaluated neutralization capacity after the fourth vaccine dose, and none has demonstrated an impact of SNA on breakthrough infection.

This study tracked antibody and cell-mediated responses in kidney transplant recipients (KTRs) receiving four doses of mRNA vaccines, comparing them with non-immunocompromised individuals undergoing dialysis (DPs). We also analyzed the anti-spike IgG subclass kinetics, which has not been previously reported in this population. Our results showed that repeated mRNA vaccinations in DPs triggered a delayed increase in IgG2 and IgG4 titers, with IgG4 being predominant, while IgG1 titers decreased after the fourth dose, like findings in healthy individuals [25]. However, in KTRs we showed that IgG1 titers increased after the second dose, lost the boosting effect after the third dose, and increased again after the fourth dose. Anti-spike IgG4 titers increased gradually after the third and fourth doses, but did not reach the high titers seen in DPs, probably due to the lack of general response shown in total IgG data.

The robust and sustained induction of anti-spike IgG4 following multiple doses of vaccine may be a consequence of ongoing class switch recombination (CSR) events in Ag-experienced B cells within germinal centers. Irrgang et al. state that the presence of anti-S IgG4 antibodies after the second immunization with the BNT162b2 vaccine indicates that a prolonged period of ongoing B cell maturation may be linked to increased CSR towards distal IgG subclasses. This, in turn, may result in the generation of IgG4-switched memory B cells over time [26]. In the present study, which encompassed four doses of mRNA-based vaccines, a comparable shift towards non-inflammatory IgG subclasses was observed. However, it should be noted that the levels and functional profile of spike-specific antibodies induced by the two vaccine types have been documented to exhibit slight disparities [27].

However, the IgG4 response is not a common result of repeated antigen exposure in vaccinations against other microorganisms, nor even of all types of SARS-CoV-2 vaccines. Several years ago, Hendrikx et al. hypothesized that the pronounced IgG4 response after some types of *Pertussis* vaccines could imply a different triggering of the immune system and a Th2 polarization of T-cells [28]. Other authors have described that IgG4 kinetics are exclusive to mRNA vaccines, not happening with other classic vaccination procedures, such as adenoviral vectors or inactivated COVID-19 vaccines [8,9,29].

It is clearly described that IgG4 and IgG2 have a lower potential to mediate FcγRs-dependent secondary effector functions, thus being considered non-inflammatory antibodies [10]. Whether the striking increase in IgG4 subclass has functional consequences for host defense mechanisms against SARS-CoV-2 infection is still under discussion. Some works have described that hybrid immunity had the highest magnitude and durability of protection [30]. In our cohort study, we observed that patients with previous infection receiving four doses of mRNA vaccine had higher SNA and more elevated anti-spike titers, specifically for the IgG4 subclass. In addition, we found evidence that increased anti-spike IgG subclass titers, including IgG4, were associated with increased SNA and a lower rate of breakthrough infection after four doses of vaccine. From these observations, it could be deduced that elevated IgG4 titers should have a positive impact on patients.

This association between anti-spike IgG4 titers and SNA was not found by Goyins et al., who investigated the antibody response to SARS-CoV-2 in a group of convalescent patients. They observed a moderate correlation between IgG1 and IgG3 and SNA, but no correlation with IgG2 and IgG4 [31].

Another remarkable finding of our study was the relationship between immunosuppressive therapy and the kinetics of anti-spike IgG subclasses. MPA had a negative effect on IgG1, IgG3, and IgG4 subclasses, whereas mTORi had a positive correlation with pro-inflammatory subclasses.

The use of MPA has been observed to have a negative effect on the efficacy of the SARS-CoV-2 vaccine in kidney transplant recipients (KTRs) and patients with autoimmune diseases, as indicated by a meta-analysis demonstrating that MPA is an independent risk factor for an adverse antibody response following immunization with two vaccine doses in KTRs [32]. In a related study, Tang et al. demonstrated that the administration of MPA to patients with systemic lupus erythematosus diminished the serological response to the SARS-CoV-2 vaccine [33], suggesting that MPA mitigated the humoral response to vaccination irrespective of the recipient’s characteristics. These findings are maintained after the fourth vaccine dose, in contrast to mTORi, which plays a protective role, as we demonstrated in our study; so much so that it is mTOR inhibitors that are mainly responsible for an adequate SNA, which decreases the risk of infection in KTRs.

The current research on mRNA vaccines has primarily focused on those targeting COVID-19, with several studies observing an increase in IgG4 antibodies associated with repeated mRNA SARS-CoV-2 vaccination and reductions in Fc-mediated antibody-dependent cellular cytotoxicity (ADCP) and antibody-dependent complement deposition (ADCD) that may limit control of viral infection [34]. This induction of IgG4 antibodies was not observed after immunization with protein vaccines or adenoviral vectors [9,29]. These data, along with our own results, set the rationale to use hybrid vaccination as the first-line protocol to induce a highly protective response in immunocompromised patients.

Finally, other kinds of vaccine against viruses other than SARS-CoV-2 have demonstrated a reduced level of seroprotection in immunocompromised populations. For instance, the influenza vaccine can generate a humoral response in transplant recipients, but with a lower level of protection, despite variable responses [35]. A less robust response has also been observed to the respiratory syncytial vaccinations in immunocompromised patients [36]. Therefore, the low response to vaccination is not unique to mRNA vaccines, although neither the response with mRNA vaccines nor the hybrid response, which, in our hands, seems to offer more robust responses, was evaluated in these studies.

This article has several limitations: First, the sample size was reduced, mainly because part of the original study population declined to receive up to four doses of the vaccine, and several of them contracted COVID-19 disease at study end points, forcing early withdrawal from the study. However, the analysis of the SARS-CoV-2 infected population has allowed us to analyze mixed immunity (by vaccination and infection) in this type of patient, as well as to determine the efficacy in terms of safety and post-infection protection of the different doses, as shown by the fact that KTRs who achieved an SNA greater than 65% after the fourth dose were almost four times less likely to become infected in the following six months.

Second, this paper focuses on the response to vaccination against the virus in its WT strain, as the vaccines administered correspond with the first waves of the pandemic (both BNT162b2 and mRNA-1273). However, in addition to the specific information regarding protection against severe SARS-CoV-2 disease, it is hoped that the data shown in this article can be extrapolated to other mRNA-based vaccines, such as versions for new variants of SARS-CoV-2 or other types of viral infections.

## 5. Conclusions

This work describes, for the first time, the dynamics of the immune response following multiple doses of mRNA-based COVID-19 vaccines in KTRs. We show that the administration of a fourth dose of mRNA vaccine significantly enhances the humoral and cellular immune response in KTRs, with a robust and sustained induction of anti-spike IgG4 that positively correlates with SNA and confers protection against breakthrough infection. Furthermore, immunosuppressive therapy has a notable impact on the kinetics of IgG subclasses, with mycophenolate having a negative effect on IgG1, IgG3, and IgG4 levels, whereas mTOR inhibitors exhibit a positive correlation with pro-inflammatory subclasses. Based on the data obtained in this article and other publications linking mTORi to improved response, we believe that non-responders may benefit from switching therapy to include an mTOR inhibitor (such as Everolimus) to increase response to vaccination, provided it is not contraindicated (proteinuria, allergy, or serious adverse effects on previous mTORi treatment).

In conclusion, these findings underscore the importance of booster vaccination strategies in combination with immunomodulatory treatments and highlight the potential of IgG4 as a key mediator of protective immunity against SARS-CoV-2 in immunocompromised populations such as KTRs.

## Figures and Tables

**Figure 1 vaccines-13-00123-f001:**
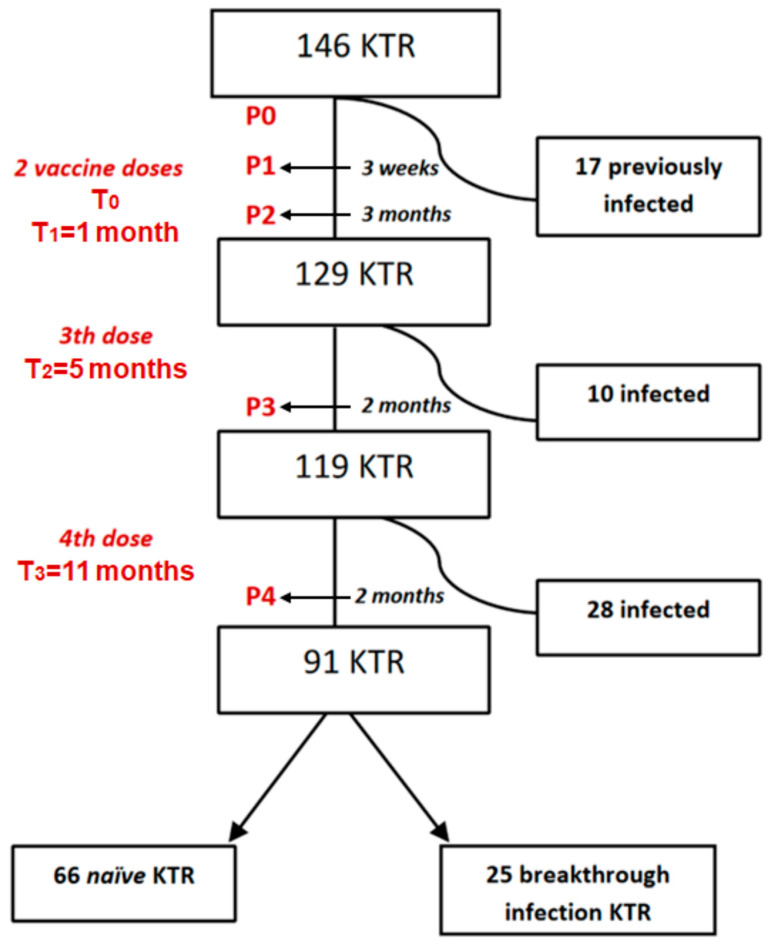
Flow chart of kidney transplant recipients (KTRs) included in the study and exclusion criteria. Each extraction (P0, P1, P2, P3, and P4) involved the following laboratory determinations: SARS-CoV-2 Serology Analysis, Assessment of Cell-Mediated Immunity, Assessment of Serum Neutralizing Activity of ACE2-RBD, and Measurement of IgG subclasses. Patients were tested for SARS-CoV-2 infection by anti-N antibodies, determined by ELISA.

**Figure 2 vaccines-13-00123-f002:**
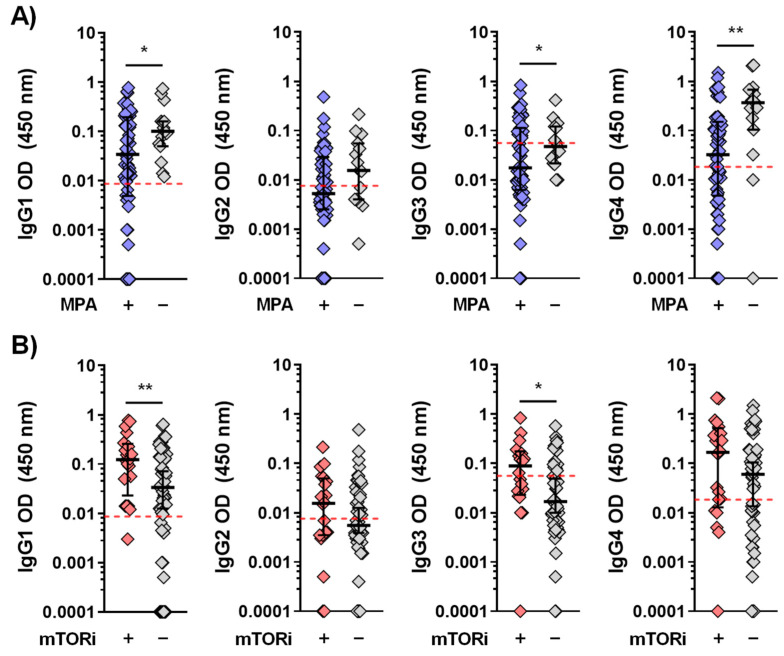
Immunosuppressive treatments modulate IgG subtypes in response to SARS-CoV-2 mRNA vaccination after booster dose. (**A**) Mycophenolate mofetil treatment (MPA) decreases IgG1 and IgG4 levels, whereas (**B**) mTOR inhibitors (mTORi) increase IgG1, IgG3, and IgG4 levels in these patients. Untreated condition refers to patients not treated with the analyzed immunosuppressant (e.g., not treated with mTORi, but with other immunosuppression). These findings highlight the distinct immunomodulatory effects of mTORi and MMF on specific IgG subclasses, suggesting possible impacts on treatment efficacy. Only significant *p*-values (<0.05) are shown, * *p*-value < 0.05, ** *p*-value < 0.01. The cut-offs were established as follows: IgG1: 0.0085; IgG2: 0.0075; IgG3: 0.0559; IgG4: 0.018.

**Figure 3 vaccines-13-00123-f003:**
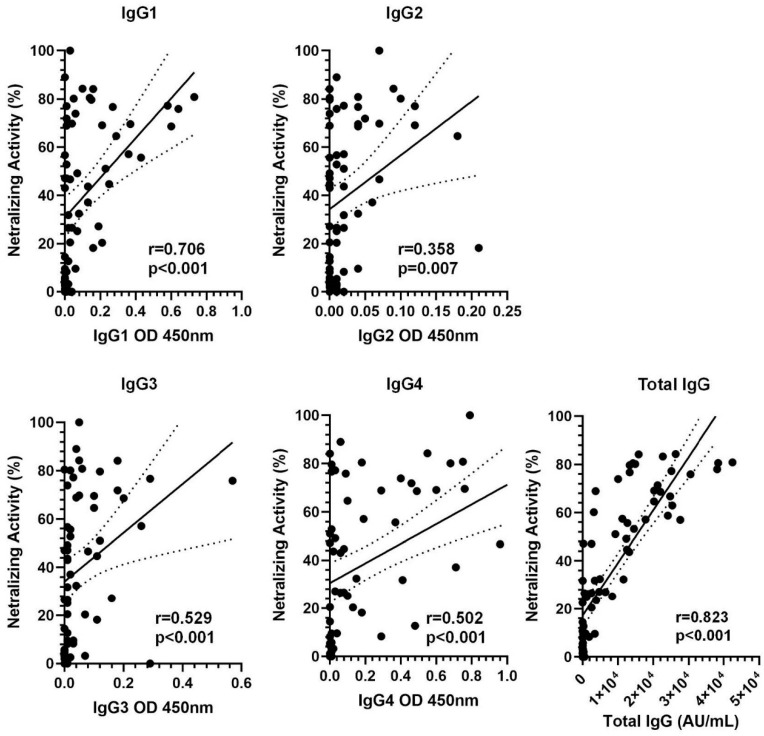
Correlation of IgG subclasses and total IgG levels with serum neutralizing activity (SNA) in kidney transplant recipients. Total IgG (r = 0.823, *p*-value < 0.001) and IgG subclasses showed a positive correlation with SNA, with IgG1 subclass having the higher r-value (0.706, *p*-value < 0.001).

**Table 1 vaccines-13-00123-t001:** Demographic characteristics and laboratory findings of kidney transplant recipients and dialysis patients.

	KTR (N = 146)	Naïve KTR (N = 91)	Infected KTR (N = 55)	DP (N = 23)
Gender (male), N (%)	91 (62.3)	55 (60.4)	36 (65.5)	13 (56.5)
Ethnicity, N (%)				
Caucasian	125 (85.6)	79 (86.8)	46 (83.6)	19 (82.6)
Hispanic	20 (13.7)	12 (13.2)	8 (14.6)	3 (13.0)
Others	1 (0.7)	0 (0)	1 (1.8)	1 (4.4)
Age				
years (mean, SD)	57 (15.0)	60.4 (15.5)	54.4 (13.5)	56 (13)
>60 y, N (%)	73 (50.0)	53 (58.2)	19 (35.2)	12 (52.1)
Diabetes, N (%)	51 (34.9)	32 (35.1)	19 (35.2)	8 (34.7)
HBP, N (%)	132 (90.4)	82 (90.1)	50 (90.9)	23 (100)
Cardiovascular disease, N (%)	34 (23.3)	21 (23.1)	13 (23.6)	8 (34.7)
Chronic lung disease, N (%)	27 (18.5)	16 (17.6)	11 (20)	5 (21.7)
Time since transplantation				
years (median, IQR)	9.2 (4.0–15.8)	9.5 (4.4–17.0)	8.0 (3.8–13.7)	
<5 years, N (%)	41 (28.1)	24 (26.4)	17 (30.9)	
Previous Transplant,				
N (%)	19 (13.0)	12 (13.2)	7 (12.7)	
Immunosuppressive drug, N (%)				
Steroids	73 (50.0)	44 (48.3)	29 (52.7)	
CNI	120 (82.2)	69 (75.8)	51 (92.7)	
MPA	111 (76.0)	70 (76.9)	41 (74.5)	
mTORi	41 (28.1)	26 (28.6)	15 (27.2)	
Thymoglobulin	82 (56.1)	53 (58.2)	29 (52.7)	
Immunosuppressive protocol, N (%)				
MPA + CNI	102 (69.8)	64 (70.3)	38 (69.1)	
MPA + mTORi	19 (13.1)	13 (14.3)	6 (10.9)	
mTORi + CNI	25 (17.1)	14 (15.4)	11 (20.0)	
eGFR (mL/min/1.73 m^2^), median (IQR)				
Stages CKD, N (%)				
>60 mL/min/1.73 m^2^	49 (33.5)	26 (28.6)	23 (41.8)	
30–60	71 (48.6)	47 (51.6)	24 (43.6)	
<30	26 (17.9)	18 (19.8)	8 (14.6)	
Cells count, 1 × 10^3^/mm^3^, median (IQR)				
Lymphocyte	1.52 (1.13–2.03)	1.54 (1.06–2.06)	1.70 (1.20–2.37)	1.62 (1.21–2.12)
CD4+ T-cells	0.54 (0.38–0.74)	0.55 (0.35–0.77)	0.59 (0.38–0.78)	
CD8+ T-cells	0.48 (0.34–1.02)	0.45 (0.29–0.78)	0.56 (0.33–0.86)	
Lymphocyte > 1 × 10^3^/mm^3^, N (%)	107 (73.2)	70 (76.9)	37 (67.3)	20 (91.3)

KTR = kidney transplant recipient. DP = dialysis patient. SD = standard deviation. IQR = interquartile ratio. CKD = chronic kidney disease. CNI = calcineurin inhibitor. MPA = mycophenolate acid. mTORi = mammalian target of rapamycin inhibitor. Egfr = estimated glomerular filtration rate.

**Table 2 vaccines-13-00123-t002:** Kinetics of IgG subclasses (median OD 450 nm [IQR]) after vaccination against SARS-CoV-2 in naïve kidney transplant recipients and dialysis patients.

	P1	P2	P3	P4
**IgG1** OD (450 nm) KTR DP	0.0043 [0.0028–0.0067] n.d.	0.0147 ^a^ [0.0044–0.0251] n.d.	0.0117 [0.0035–0.1069] 0.0490 [0.0290–0.0845]	0.0470 ^b^ [0.0103–0.1669] 0.0170 ^c^ [0.0065–0.0320]
**IgG2** OD (450 nm) KTR DP	0.0020 [0.0010–0.0035] n.d.	0.0083 ^a^ [0.005–0.0183] n.d.	0.0095 [0.0038–0.0226] 0.0260 [0.0218–0.0495]	0.0062 [0.0034–0.0214] 0.0530 [0.0123–0.0754]
**IgG3** OD (450 nm) KTR DP	0.0227 [0.0108–0.03725] n.d.	0.0323 ^d^ [0.0115–0.0703] n.d.	0.0215 [0.0105–0.121] 0.0330 [0.0120–0.0470]	0.0160 [0.007–0.0775] 0.0415 [0.0284–0.0641]
**IgG4** OD (450 nm) KTR DP	0.0060 [0.0010–0.0093] n.d.	0.0052 [0.0024–0.011] n.d.	0.0070 ^e^ [0.0019–0.0371] 0.7808 [0.0363–1.1873]	0.0285 ^f^ [0.0048–0.1860] 0.8755 [0.1826–1.4783]

P1: At 21 days after 2nd dose. P2: At 3 months after 2nd dose. P3: At 2 months after 3th dose. P4: At 2 months after 4th dose. KTR: kidney transplant recipient (n = 60). DP: dialysis patient (n = 12). ^a^ IgG1, IgG2 at P1 vs. P2, *p* < 0.001; ^b^ IgG1 in KTRs at P3 vs. P4, *p* = 0.010; ^c^ IgG1 in DP at P3 vs. P4, *p* = 0.005; **^d^** IgG3 at P1 vs. P2, *p* = 0.016; **^e^** IgG4 in KTRs at P2 vs. P3, *p* = 0.007; ^f^ IgG4 in KTRs at P3 vs. P4 *p* = 0.001. The cut-offs were established as follows: IgG1: 0.0085; IgG2: 0.0075; IgG3: 0.0559; IgG4: 0.018.

**Table 3 vaccines-13-00123-t003:** Serum neutralizing activity >65% (percentile 75) after 4 doses of vaccine in naïve KTRs.

SNA > 65%	Univariate		Multivariate	
	OR (CI 95%)	*p*	OR (CI 95%)	*p*
Diabetes	1.81 (0.64–5.07)	0.200		
Age >60 years	0.67 (0.48–0.95)	0.077	0.49 (0.15–1.55)	0.227
Gender male	0.90 (0.58–1.39)	0.671		
Steroids	0.54 (0.36–0.80)	0.019	0.83 (0.24–2.77)	0.764
CNI	0.98 (0.71–1.36)	0.939		
MPA	0.89 (0.69–1.16)	0.490	0.52 (0.13–2.03)	0.350
mTORi	5.20 (0.77–35.17)	0.027	**4.29 (1.21–15.17)**	**0.024**
Thymoglobulin	0.67 (0.47–0.95)	0.073	0.92 (0.25–3.30)	0.902
eGFR > 30 mL/min	3.03 (0.87–10.55)	0.072	2.89 (0.31–26.68)	0.349
<5 years since transplantation	0.93 (0.68–1.26)	0.685		
Previous transplant	1.85 (0.49–6.87)	0.353		
Lymphocyte > 1 × 10^3^/mm^3^	0.97 (0.35–2.63)	0.953		

CNI = calcineurin inhibitor. MPA = mycophenolate acid. mTORi = mammalian target of rapamycin inhibitor. eGFR = estimated glomerular filtration rate.

**Table 4 vaccines-13-00123-t004:** Characteristics of specific immunity 2 months after the fourth dose (P4) according to previous SARS-CoV-2 infection.

SARS-CoV-2 Infection Before 4th Dose	Naïve KTR (N = 91)	Infected KTR (N = 55)	*p*
IgG SARS-CoV-2 (BAU/mL)	516.9 (39.6–2263.5)	3389.1 (1707.6–8427.1)	**<0.001**
SNA (%)	30.0 (7.9–62.9)	74.4 (51.3–82.4)	**<0.001**
IgG1 (OD 450 nm)	0.047 (0.006–0.193)	0.078 (0.017–0.176)	0.450
IgG2 (OD 450 nm)	0.006 (0.002–0.022)	0.026 (0.003–0.056)	0.241
IgG3 (OD 450 nm)	0.016 (0.006–0.077)	0.098 (0.055–0.222)	**<0.001**
IgG4 (OD 450 nm)	0.028 (0.003–0.214)	0.155 (0.056–0.912)	**0.003**
IFNy-CD4 (%)	0.28 (0.02–0.56)	0.31 (0.09–0.55)	0.608
IFNy-CD8 (%)	0.15 (0.01–0.48)	0.30 (0.15–0.83)	**0.005**

KTR = kidney transplant recipient. SNA = serum neutralizing activity. IFNy = interferon gamma. OD = optical density.

**Table 5 vaccines-13-00123-t005:** Characteristics of specific immunity at P4 between naïve and infected KTRs at nine months after the fourth dose of the vaccine.

SARS-CoV-2 Infection After 4th Dose	Naïve KTR (N = 66)	Infected KTR (N = 25)	*p*
IgG SARS-CoV-2 (BAU/mL)	1429.7 (63.8–2870.4)	121.6 (1.0–869.6)	**0.005**
SNA (%)	45.8 (8.5–69.4)	20.3 (1.9–46.1)	**0.012**
IgG1 (OD 450 mm)	0.050 (0.006–0.205)	0.039 (0.000–0.060)	**0.009**
IgG2 (OD 450 mm)	0.005 (0.002–0.037)	0.002 (0.000–0.006)	**0.006**
IgG3 (OD 450 mm)	0.017 (0.006–0.095)	0.000 (0.000–0.014)	**<0.001**
IgG4 (OD 450 mm)	0.032 (0.003–0.367)	0.001 (0.000–0.012)	**<0.001**
IFNy-CD4 (%)	0.28 (0.00–0.56)	0.30 (0.09–0.64)	0.373
IFNy-CD8 (%)	0.17 (0.00–0.68)	0.12 (0.00–0.46)	0.421

KTR = kidney transplant recipient. SNA = serum neutralizing activity. IFNy = interferon gamma. OD = optical density.

## Data Availability

The original contributions presented in the study are included in the article. Further inquiries can be directed to the corresponding author.

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
