# Peer review of "Immunosuppressive Therapy Modifies Anti-Spike IgG Subclasses Distribution After Four Doses of mRNA Vaccination in a Cohort of Kidney Transplant Recipients"

_vaccines, 2025, doi:10.3390/vaccines13020123_

Round 1

Reviewer 1 Report

Comments and Suggestions for Authors

Introduction:
The introduction provides sufficient background and sets the context well. However, the relevance of IgG subclass kinetics in kidney transplant recipients (KTR) could be emphasized more explicitly in the broader context of vaccine development for immunosuppressed populations.

Research Design:
While the research design is appropriate, the authors should consider including more detailed demographic variability (e.g., ethnicity, comorbidities) in their control and patient groups to enhance generalizability. Additionally, expanding the study to include other SARS-CoV-2 variants, such as Omicron, would improve its applicability to current public health concerns.

Methods:
The methods section is detailed and well-structured. However, the authors should:

  • Provide a clearer justification for the cutoff values used in the ELISA assays.
  • Explain the rationale behind the chosen timeline for vaccination and sampling intervals.
  • Add a concise visual flowchart summarizing the experimental workflow to aid reader comprehension.

Results Presentation:
The results are generally well-presented. Nonetheless:

  • Figures such as Figure 2 could benefit from improved labeling and clearer legends for easier interpretation.
  • Including confidence intervals for key outcomes in tables would enhance statistical clarity.

Discussion and Interpretation:
The discussion effectively highlights the implications of the findings. To improve:

  • Address the functional significance of IgG4 and its potential anti-inflammatory role in greater depth.
  • Include a brief comparison of the findings with data on other immunocompromised populations or different vaccine platforms.
  • Expand the acknowledgment of limitations, especially regarding the study's focus on the WT SARS-CoV-2 strain and the relatively small control group size.

Clinical Implications and Recommendations:
The manuscript could include more explicit recommendations for clinical practice based on the findings. For instance, the potential of mTOR inhibitors (mTORi) as an adjunct therapy to enhance vaccine responses should be elaborated upon.

References:
The references are relevant and appropriate. However, consider including additional citations for emerging research on IgG4 dynamics in response to mRNA vaccines in different populations.

Comments on the Quality of English Language

The quality of the English language in the manuscript is generally good, with well-structured sentences and a clear presentation of ideas. However, there are a few areas that could be improved:

  1. Clarity and Readability:

    • Some sentences are overly complex and could benefit from simplification for better readability.
    • Example: "These findings highlight the distinct immunomodulatory effects of mTORi and MMF on specific IgG subclasses, suggesting possible impacts on treatment efficacy."
      Suggested Revision: "These findings underscore how mTORi and MMF differently modulate IgG subclasses, potentially affecting treatment outcomes."
  2. Grammar and Word Choice:

    • Minor grammatical errors or awkward word choices are present, such as the inconsistent use of tenses.
    • Example: "The cut-offs were stables as follows"
      Suggested Revision: "The cut-offs were stable as follows."
  3. Terminology:

    • While the terminology is appropriate for a scientific audience, some technical terms (e.g., "serum neutralizing activity," "FcγRs") may benefit from brief definitions or clarifications for non-expert readers.
  4. Flow and Transition:

    • Transitions between sections could be smoother. For example, the shift from describing results to discussing implications feels abrupt in some places.
  5. Typographical Errors:

    • There are a few typographical inconsistencies (e.g., spacing in tables, alignment of figure captions).
  6. Formal Tone:

    • The tone is appropriately formal and scientific, but occasional phrases could be more concise.

Author Response

Q1)

Introduction:
The introduction provides sufficient background and sets the context well. However, the relevance of IgG subclass kinetics in kidney transplant recipients (KTR) could be emphasized more explicitly in the broader context of vaccine development for immunosuppressed populations.

A1) As other solid organ transplant recipients (SOTRs), the IgG subclass response to vaccines, such as the COVID-19 mRNA vaccines, mainly include IgG1, IgG3, and IgG4. This profile is similar to several results from studies that focuses on healthy controls, although the overall antibody response is lower in SOTRs, as we also described previously in a work with total IgG (Lee J 2023 & 2024)

  1. Lee, J., Shapiro, J., Lee, J., Sitaris, I., Peralta, S., Li, M., Wouters, C., Karaba, A., Blankson, J., Tobian, A., Werbel, W., Pekosz, A., & Klein, S. (2023). Altered antibody profiles after receipt of third dose SARS-CoV-2 mRNA vaccines in solid organ transplant recipients. The Journal of Immunology. https://doi.org/10.4049/jimmunol.210.supp.252.16.
  2. Lee, J., Sachithanandham, J., Lee, J., Shapiro, J., Li, M., Sitaris, I., Peralta, S., Wouters, C., Cox, A., Segev, D., Durand, C., Robien, M., Tobian, A., Karaba, A., Blankson, J., Werbel, W., Pekosz, A., & Klein, S. (2024). A Third COVID-19 Vaccine Dose in Kidney Transplant Recipients Induces Antibody Response to Vaccine and Omicron Variants but Shows Limited Ig Subclass Switching. bioRxiv. https://doi.org/10.1101/2024.09.01.610689.

Q2)

Research Design:

While the research design is appropriate, the authors should consider including more detailed demographic variability (e.g., ethnicity, comorbidities) in their control and patient groups to enhance generalizability. Additionally, expanding the study to include other SARS-CoV-2 variants, such as Omicron, would improve its applicability to current public health concerns.

A2) We have included a more detailed demographic information about the patients in the Table 1.

Methods:
The methods section is detailed and well-structured. However, the authors should:

Q3) Provide a clearer justification for the cutoff values used in the ELISA assays.

A3) Cut-off establishment: To establish whether a sample could be deemed as positive, an OD threshold was obtained for every IgG subclass, as previously described in an in-house ELISA assay [Rodriguez-Juan C 2003 and Pérez-Flores 2023]. The threshold was established from the mean OD and standard deviation (SD) values achieved with the pre-pandemic samples as MeanOD + 2 * SD (CI 95%). Individuals with an OD value above the cut-off point were considered positive.

We included a sentence explaining the methodology to establish the cut-offs in the manuscript (lines 165-172), reading:

“To establish the cut-off of anti-S IgG subclasses, we used the value of the mean plus twice the standard deviation (MeanOD + 2 * SD , 95% CI) of the OD value at 430nm of 8 pre-pandemic sera (PCR negative, anti-S and anti-N total IgG negative and with no COVID-19 compatible symptoms) per ELISA plate. This methodology is based on the fact that 95.0% of the data are within 1.96 standard deviations of the mean. Therefore, values outside this range are unlikely to be due to random sampling and are significantly higher than the cut-off point value obtained.”

  1. Rodríguez-Juan C., Sala-Silveira L., Pérez-Blas M., Valeri A.P., Aguilera N., López-Santalla M., Fuertes A., Martín-Villa J.M. Increased levels of bovine serum albumin antibodies in patients with type 1 diabetes and celiac disease-related antibodies. Pediatr. Gastroenterol. Nutr. 2003;37:132–135. doi: 10.1097/00005176-200308000-00009.
  2. Pérez-Flores, Isabel et al. “Role of mTOR inhibitor in the cellular and humoral immune response to a booster dose of SARS-CoV-2 mRNA-1273 vaccine in kidney transplant recipients.” Frontiers in immunology 14 1111569. 2 Feb. 2023, doi:10.3389/fimmu.2023.1111569

Q4) Explain the rationale behind the chosen timeline for vaccination and sampling intervals.

A4) The vaccination periods were carried out following the schedules and periods pro-vided by the Spanish Ministry of Health, including the administration of a booster (4th dose) of the vaccination in individuals undergoing renal transplantation. The vaccination protocol was the same for all participants, including 3 doses of mRNA-1273 vaccine and a fourth dose of BNT162b2 vaccine (due to stock limitations).

Samples were taken according to the literature available at the time, although a standard of two months after the booster dose was established for the last samples, based on the results published after one year of analysis of immune responses in the general population

We have included this information in the manuscript, lines 119-127.

Q5) Add a concise visual flowchart summarizing the experimental workflow to aid reader comprehension.

A5) We have updated the Figure 1 (Flowchart) now showing a more precise experimental workflow to ease the comprehension of the methodology and the time when samples were analyzed. We also updated the figure legend to improve the comprehension of the flowchart (lines 102-107).

Results Presentation:

The results are generally well-presented. Nonetheless:

Q6) Figures such as Figure 2 could benefit from improved labeling and clearer legends for easier interpretation.

A6) We have improved the labeling and legend from most of the figures, as well as we included visual representation of the tables to easy the visualization and interpretation of the results.

Q7) Including confidence intervals for key outcomes in tables would enhance statistical clarity.

A7) We included the interquartile range (IQR) in the tables where it was absent (see Table 2.) We also included in the figure legend that the IQR is included below the data.

As suggested by the reviewer 3, we also included a Supplementary Figure S2, showing a graphical visualization off the antibody trajectories and comparison between KTR and controls for the different IgG subclasses of the samples P3 and P4 (as P1 and P2 only have the data from patients).

Q8) Discussion and Interpretation:

The discussion effectively highlights the implications of the findings. To improve:

  • Address the functional significance of IgG4 and its potential anti-inflammatory role in greater depth.
  • Include a brief comparison of the findings with data on other immunocompromised populations or different vaccine platforms.

A8) We have included new references, comparisons and discussion about both IgG4 functional significance and data from other populations and different vaccine platforms in the discussion section (lines 377-392).

Q9) Expand the acknowledgment of limitations, especially regarding the study's focus on the WT SARS-CoV-2 strain and the relatively small control group size.

A9) We expand the acknowledgment of limitations in the corresponding section, now reading:

This article has several of limitations: First, the sample size has been reduced, mainly because part of the original study population declined to receive up to four doses of the vaccine, and several of them contracted COVID-19 disease at study end points, forcing early withdrawal from the study. However, the analysis of the SARS-CoV-2 infected population has allowed us to analyze mixed immunity (by vaccination and infection) in this type of patients, as well as to determine the efficacy in terms of safety and post-infection protection of the different doses, as shown by the fact that KTR who achieved an SNA greater than 65% after the fourth dose were almost four times less likely to become infected in the following six months.

Second, the paper focuses on the response to vaccination against the virus in its WT strain, as the vaccines administered correspond with the first waves of the pandemic (both BNT162b2 and mRNA-1273). However, in addition to the specific information regarding protection against severe SARS-CoV-2 disease, it is hoped that the data shown in this article can be extrapolated to other mRNA-based vaccines, such as versions for new variants of SARS-CoV-2 or other types of viral infections.

(lines 394-408)

Q10) Clinical Implications and Recommendations:

The manuscript could include more explicit recommendations for clinical practice based on the findings. For instance, the potential of mTOR inhibitors (mTORi) as an adjunct therapy to enhance vaccine responses should be elaborated upon.

A10) We agree with the reviewer that our results could be beneficial to improve the immune response in immunosuppressed patients, like KTR our with other kind of solid transplants. We have updated the Conclusions Section to include more explicit recommendation for clinical practice based on the results of this work (lines 412-428).

Q11) References:

The references are relevant and appropriate. However, consider including additional citations for emerging research on IgG4 dynamics in response to mRNA vaccines in different populations.

A11) We have included new references about IgG4 dynamics in response to mRNA vaccines in different populations, as well as the effect of non-mRNA vaccines in the IgG1-4 subclasses kinetics (36-38) with their corresponding new sentences in the manuscript (see Discussion section).

Reviewer 2 Report

Comments and Suggestions for Authors

Question regarding testing for SARS-CoV-2 infection?

- Was there a testing protocol?

- Which test was used?

-If there was no testing protocol- how were the asymptomatic subjects diagnosed?

Author Response

Q1) Question regarding testing for SARS-CoV-2 infection?

- Was there a testing protocol?

- Which test was used?

-If there was no testing protocol- how were the asymptomatic subjects diagnosed?

A1) We thank the reviewer for its positive comments and suggestions.

We detected the subjects that suffered SARS-CoV-2 infection prior to vaccination or in periods between sample collections by analyzing the presence of anti-Nucleoprotein (N) antibodies employing an ELISA assay. As we describe in our previously published work (Pérez-Flores I, Juarez I, et al. 2023. Front Immunol). Briefly, 96-well flat-bottom plates were coated with 2 μg/mL SARS-CoV-2 N-protein and 1:100 dilutions of the sera were incubated for 30 minutes at room temperature (RT), washed 5 times and detected with a goat anti-human IgG HRP-conjugated antibody (ThermoFisher Scientific). ELISA was developed with TMB and HCl and measured at 430 nm. To establish the cut-off of anti-N antibodies, we used the value of the mean plus twice the standard deviation (95% CI) of the absorbance value at 430nm of 8 pre-pandemic sera (PCR negative, anti-S IgG negative and with no COVID-19 compatible symptoms) per ELISA plate.

We included this information in the manuscript, lines 137-139, reading:

“To identify individuals who had been infected with the SARS-CoV-2 before vaccination or between sample collections, we employed an ELISA assay that assessed the presence of anti-SARS-CoV-2 N-protein antibodies as previously described 5.”

Reviewer 3 Report

Comments and Suggestions for Authors

This manuscript investigates the IgG subclass profile and T-cell response in a cohort of kidney transplant recipients. The findings are both informative and insightful.

Major Concerns
1. Kinetics and Antibody trajectory visualisation:
The manuscript frequently references the kinetics of IgG subclasses, but the associated analyses, outcomes, and visualisations are underdeveloped.
Lines 186-202 present only ODâ‚„â‚…â‚€ outcomes, which could benefit from more detailed visual representation.

I recommend creating visualisations of antibody trajectories to provide a comprehensive overview of immunity within this cohort. To enhance data presentation, consider total participants and subgroup analyses (e.g., IgG total and subclasses, IFN-γ levels, drug treatments, naïve vs. hybrid immunity) in separate figures (e.g., Fig Xb, Xc).

For reference, please take a look at this example: Antibody Trajectory Visualisation. https://www.ijidonline.com/cms/10.1016/j.ijid.2024.107111/asset/52b375ee-c18b-4857-8159-8d9069111fbb/main.assets/gr1_lrg.jpg  

Comments
1. SARS-CoV-2 IgG II Quant standardisation unit (BAU/mL):
The test results are standardised to BAU/mL using a conversion factor of 0.142 (BAU/mL = AU/mL × 0.142).
Recommend reporting outcomes in BAU/mL for easier comparison with standardised tests from other kits. Additionally, add a statement about the conversion factor in lines 121-127.
Reference, Table 1: https://pmc.ncbi.nlm.nih.gov/articles/PMC9388276

2. Subgroup analysis by vaccine Use:
If data allow, suggest subgrouping participants by vaccine regimens, such as homologous BNT162b2, homologous mRNA-1273, and heterologous mRNA combinations.
Each vaccine differs in formulation, mRNA sequences, and dosage (e.g., BNT162b2 30 μg, mRNA-1273 100 μg for primary doses, and 50 or 100 μg for boosters).
These differences could yield insightful outcomes through subgroup analyses.

Typos/Terms
1. Consistency in naming:
I suggest using "Pfizer—BioNTech" to align with the standard terminology in most literature.

2. Terminology:
Replace "IgG SARS-CoV-2 (UA/ml)" with the appropriate standardised term.

Author Response

Comments and Suggestions for Authors

This manuscript investigates the IgG subclass profile and T-cell response in a cohort of kidney transplant recipients. The findings are both informative and insightful.

Major Concerns

Q1. Kinetics and Antibody trajectory visualisation:

The manuscript frequently references the kinetics of IgG subclasses, but the associated analyses, outcomes, and visualisations are underdeveloped.

Lines 186-202 present only ODâ‚„â‚…â‚€ outcomes, which could benefit from more detailed visual representation.

A1) As subjected by the reviewer, we developed a visual representation (Supplementary Figure S2) to allow the comparison between KTR and the control group, that clearly show the differences in the kinetics of the IgG subclasses after the third and fourth doses.

Q2) I recommend creating visualisations of antibody trajectories to provide a comprehensive overview of immunity within this cohort. To enhance data presentation, consider total participants and subgroup analyses (e.g., IgG total and subclasses, IFN-γ levels, drug treatments, naïve vs. hybrid immunity) in separate figures (e.g., Fig Xb, Xc).

For reference, please take a look at this example: Antibody Trajectory Visualisation. https://www.ijidonline.com/cms/10.1016/j.ijid.2024.107111/asset/52b375ee-c18b-4857-8159-8d9069111fbb/main.assets/gr1_lrg.jpg  

A2) We included a Supplementary Figure S3 to show the trajectories of immunity (total IgG and IgG1-4 subclasses) in KTR, with subgroup analyses that include humoral data of the whole group of KTR, mTORi and MMF and naïve vs hybrid immunity between the 3rd and 4th doses, as the previous ones are already available in a previous work from our group (see cite 5, Pérez-flores I, Juarez I et al. 2023 Front Immunol). We also included the figure legend for the new graphs.

Comments
1. SARS-CoV-2 IgG II Quant standardisation unit (BAU/mL):

The test results are standardised to BAU/mL using a conversion factor of 0.142 (BAU/mL = AU/mL × 0.142).
Q3) Recommend reporting outcomes in BAU/mL for easier comparison with standardised tests from other kits. Additionally, add a statement about the conversion factor in lines 121-127.
Reference, Table 1: https://pmc.ncbi.nlm.nih.gov/articles/PMC9388276

A3) We have standardized IgG results to BAU/mL as requested, and we included the statement about conversion:

“The antibody values were converted into WHO binding antibody units (BAU)/ml using a conversion factor of 0.142 (BAU/mL = AU/mL × 0.142).” (Lines 141-142)

  1. Subgroup analysis by vaccine Use:

Q4) If data allow, suggest subgrouping participants by vaccine regimens, such as homologous BNT162b2, homologous mRNA-1273, and heterologous mRNA combinations.
Each vaccine differs in formulation, mRNA sequences, and dosage (e.g., BNT162b2 30 μg, mRNA-1273 100 μg for primary doses, and 50 or 100 μg for boosters).
These differences could yield insightful outcomes through subgroup analyses.

A4) All the patients analyzed received the same vaccination schedule and number of doses: Three mRNA-1273 and the 4th dose of BNT162b2, which make impossible to analyze the different outcomes of patients vaccinates with one or the other type of mRNA-based vaccine.

We have included this point in the “Methods – Patient Selection”, lines 119-123.

Typos/Terms
1. Consistency in naming:

I suggest using "Pfizer—BioNTech" to align with the standard terminology in most literature.

Answer: This has been corrected

  1. Terminology:

Replace "IgG SARS-CoV-2 (UA/ml)" with the appropriate standardised term.

Answer: This has been corrected

Round 2

Reviewer 3 Report

Comments and Suggestions for Authors

Suggest clarifying the dosage of the mRNA-1273 booster (third dose) in the manuscript (lines 121-123).
The manufacturer recommends a dose of 50 μg (0.25 mL) per shot; however, certain studies or specific target groups, particularly immunocompromised individuals, may use a 100 μg dose, identical to the primary series. Providing this information will help ensure clarity and prevent potential misunderstandings regarding the dosing protocol.

Author Response

Q1) Suggest clarifying the dosage of the mRNA-1273 booster (third dose) in the manuscript (lines 121-123).
The manufacturer recommends a dose of 50 μg (0.25 mL) per shot; however, certain studies or specific target groups, particularly immunocompromised individuals, may use a 100 μg dose, identical to the primary series. Providing this information will help ensure clarity and prevent potential misunderstandings regarding the dosing protocol.

A1) We have included the information of the booster (third) dose following the reviewer's suggestion. Now reads:

"The vaccination periods were carried out following the schedules and periods provided by the Spanish Ministry of Health, including the administration of a booster (4th dose) of the vaccination in individuals undergoing renal transplantation. The vaccination protocol was the same for all participants, including 2 initial doses of mRNA-1273-, a booster (third dose) of mRNA1237 was performed with 0.25 ml (containing 50 μg of mRNA, i.e., half the dose of the initial regimen and a fourth dose of BNT162b2 vaccine (due to stock limitations)."

Lines 119-125